# Developing a Low-Cost Device for Estimating Air–Water ΔpCO_2_ in Coastal Environments

**DOI:** 10.3390/s25113547

**Published:** 2025-06-04

**Authors:** Elizabeth B. Farquhar, Philip J. Bresnahan, Michael Tydings, Jessie C. Jarvis, Robert F. Whitehead, Dan Portelli

**Affiliations:** 1Department of Earth and Ocean Sciences, University of North Carolina Wilmington, Wilmington, NC 28403, USA; 2Center for Marine Science, University of North Carolina Wilmington, Wilmington, NC 28409, USA; 3Department of Biology and Marine Biology, University of North Carolina Wilmington, Wilmington, NC 28403, USA

**Keywords:** low cost, sensor, CO_2_, coastal, pCO_2_, delta pCO_2_, instrumentation, microcontroller

## Abstract

The ocean is one of the world’s largest anthropogenic carbon dioxide (CO_2_) sinks, but closing the carbon budget is logistically difficult and expensive, and uncertainties in carbon fluxes and reservoirs remain. One specific challenge is that measuring the CO_2_ flux at the air–sea interface usually requires costly sensors or analyzers (USD > 30,000), which can limit observational capacity. Our group has developed and validated a low-cost ΔpCO_2_ system, able to measure both pCO_2_^water^ and pCO_2_^air^, for USD ~1400 to combat this limitation. The device is equipped with Internet of Things (IoT) capabilities and built around a USD ~100 pCO_2_ K30 sensor at its core. Our Sensor for the Exchange of Atmospheric CO_2_ with Water (SEACOW) may be placed in an observational network with traditional pCO_2_ sensors or ∆pCO_2_ sensors to extend the spatial coverage and resolution of monitoring systems. After calibration, the SEACOW reports atmospheric pCO_2_ measurements that are within 2–3% of the measurements made with a calibrated LI-COR LI-850. We also demonstrate the SEACOW’s ability to capture diel pCO_2_ cycling in seagrass, provide recommendations for SEACOW field deployments, and provide additional technical specifications for the SEACOW and for the K30 itself (e.g., air- and water-side 99.3% response time; 5.7 and 29.6 min, respectively).

## 1. Introduction

The ocean absorbs approximately 26% of the CO_2_ emitted from human activities every year [1], demonstrating its critical role in buffering climate change. However, the ocean’s ability to store carbon varies significantly based on temperature, habitat type, circulation patterns, organic carbon concentration, alkalinity, and more, exemplifying the complex nature of the carbon cycle. For instance, coastal environments with upwelling may be net sources and outgas CO_2_ [2], while some habitats are net sinks, sequestering CO_2_ [3]. Moreover, there is a large degree of temporal variability in carbon cycling in aquatic ecosystems, with some switching from a net sink to a net source or vice versa throughout the year [4,5]. In a study examining global estimates of CO_2_ fluxes, the authors were able to demonstrate 15% more variance in coastal regions by increasing the resolution of their observations from 1° × 1° by 1 month to 0.25° × 0.25° by 8 days [6]. Therefore, habitat monitoring at a high spatial and temporal resolution is crucial to understanding variable carbon budgets. Air–water CO_2_ fluxes are one piece of the carbon budget that may offer insight into a given habitat’s role in the ocean carbon cycle.

Seagrass meadows are one type of “blue carbon” ecosystem especially well studied for their role in carbon cycling. Despite only covering 0.2% of the ocean’s surface, seagrass may be responsible for 10% of the organic carbon stored in the ocean [7,8]. However, these estimates were made using data compiled from around the globe, which can lack the spatial resolution needed to advise local management decisions [8,9]. Furthermore, modeling general coastal CO_2_ uptake rates can be difficult, with some models overestimating or underestimating the rates [10,11], demonstrating the need for more direct measurements of the environments to inform such models.

Currently, air–water CO_2_ fluxes are calculated from measurements using floating chambers, ΔpCO_2_ devices (able to measure both pCO_2_^water^ and pCO_2_^air^) (CO2-Pro ATM Pro-Oceanus, Canada; 32,000 USD), aquatic pCO_2_ devices that assume a spatially uniform atmospheric CO_2_ [12], or eddy covariance instrumentation (LI-COR, Lincoln, NE, USA; 26,500 USD) [13]. With high-quality (and typically higher-cost) devices or discrete sampling methodologies, it can be difficult to collect enough data to thoroughly resolve spatial variability. As a result, several groups have developed low-cost aquatic pCO_2_ sensors, whose values can contribute to the estimation of CO_2_ fluxes according to the following Equation (1):(1)$$F=(k_{w}*K_{H}*∆pCO_{2})$$
where $k_{w}$ is the gas transfer velocity (m hr^−1^), $K_{H}$ is the solubility of CO_2_ (mol m^−3^ atm^−1^), which depends on pressure, temperature, and salinity, and $∆pCO_{2}$ is (pCO_2_^water^–pCO_2_^air^) in atm [14,15]. The $∆pCO_{2}$ value and, subsequently, F will be positive if pCO_2_^water^ > pCO_2_^air^. Subsequently, F will be negative when pCO_2_^water^ < pCO_2_^air^. Several low-cost aquatic pCO_2_ sensor examples include the SIP-CO2 [16], the ACDC [17], the Gas-Pro [18], and a fluorescent pCO_2_ sensor [19]. Additionally, the Fluxbot was developed as a low-cost CO_2_ flux chamber, and although it is intended for terrestrial use, it is a notable example of low-cost CO_2_ flux technology [20]. Here, we use USD 5000 in 2024 as a loose threshold for low cost. There are also several mid- to high-range commercial pCO_2_ sensors, including, for example, the Sunburst SAMI-CO2, Turner Designs C-Sense, the Pro-Oceanus CO_2_-Pro CV, the Pro-Oceanus Mini CO2, and the CONTROS HydroCR CO_2_ (USD 7000–20,000 at the time of writing). Despite the availability of several pCO_2_ sensor/analyzer options, we are not aware of other low-cost ∆pCO_2_ devices. Several studies have built ∆pCO_2_ systems using more expensive sensors and, frequently, onboard standards for autonomous calibration [21,22,23]. A ∆pCO_2_ device is especially advantageous because drift, or the gradual decrease in sensor accuracy, can theoretically be minimized, which is one of the greatest challenges with low-cost instrumentation. When taking the difference between pCO_2_^water^ and pCO_2_^air^, the drift that has occurred on the sensor is subtracted out of the final ∆pCO_2_, providing a more robust value (assuming that drift occurs as in the form of an offset that affects both water- and air-side measurements equally).

In this paper, we detail the creation of a low-cost Internet of Things (IoT) ∆pCO_2_ device, termed the SEACOW, or System for the Exchange of Atmospheric CO_2_ with Water, built for USD ~1400. We conducted laboratory and field investigations to characterize the SEACOW and its K30 CO_2_ sensor (e.g., response time, accuracy, power budget, and deployment length), as well as examined its ability to capture the diel cycling of pCO_2_ due to seagrass productivity. While we primarily discuss its potential use in coastal environments, the SEACOW may be also be deployed in freshwater systems.

## 2. Materials and Methods

### 2.1. Developing the SEACOW

#### 2.1.1. Internal Components

The SEACOW is designed to sit at the surface of the water, half submerged, and with one CO_2_ exchanger in the air and one in the water. CO_2_ permeates through expanded PTFE (ePTFE) membranes, enters an internal, sealed air stream, and flows through a solenoid valve, 61 cm of Nafion tubing placed inside ~160 g of Drierite desiccant, a gas pump, a 35 µm filter, into a custom K30 housing, and finally to a second 3-way solenoid valve (Figure 1). The solenoid valves are programmed to switch the air flow between the air-side exchanger and the water-side exchanger, depending on which side is being sampled at that time. Tubing was used for the water side of the SEACOW to decrease bubbles that could get trapped on the surface of the planar ePTFE when deployed.

#### 2.1.2. Electronics and Software

The SEACOW hardware is composed of 11 electrical components, which are detailed in a circuit diagram with part numbers in Appendix B. However, the primary electrical components include the Boron microcontroller (Particle, San Francisco, CA, USA), the Adalogger Featherwing RTC+SD board (Adafruit, NYC, New York, NY, USA), and the K30 sensor (Senseair, Delsbo, Sweden). The Boron is the “brain” of the SEACOW, as it carries out the source code to control all other electrical components and powers the rest of the devices. The Boron is powered via a 3.7 V LiPo battery and can be communicated with via a micro-USB cable to update firmware. The source code for the Boron was developed using the Particle Workbench extension in Visual Studio Code version 1.95.3 and Particle Device OS 2.3.0.

The K30 is a non-dispersive infrared (NDIR) CO_2_ sensor (USD 99) with an accuracy of ± 30 µatm or ± 3% of the reading (whichever is greater), which can be improved significantly with calibration [16,17,25,26]. While the K30 manual states it measures XCO_2_, it was empirically determined by Wall (2014) that the K30 measures pCO_2_, with which our group concurred after preliminary testing (in other words, the readings are proportional to pressure) [17]. Additionally, the K30 has an automatic baseline correction (ABC) algorithm that helps to deter long-term drift by assuming the lowest concentration measured in the last 7.5 days is the atmospheric level and corrects itself by shifting its readings. Therefore, we turned off the ABC algorithm prior to deployments using methods described in the K30 Modbus manual, since we were not measuring atmospheric values only, and therefore, lower values are possible [27]. We encased the K30, which has multiple gas inlets and outlets into the optical sensing path, in a custom 3D-printed housing and sealed it with marine epoxy to ensure airtightness and to control flow into and out of the sensor (Figure 2). Additionally, we placed a BME280 Protoboard (Adafruit, NYC, USA) inside the K30 housing to measure temperature, pressure, and humidity. This protoboard uses the BME280 sensor manufactured by Bosch Sensortec of Germany. The BME280 reads humidity with ± 3% accuracy, barometric pressure with ± 1 hPa absolute accuracy, and temperature with ± 1.0 °C accuracy. All materials used to build the SEACOW are listed on our GitHub repository (V1.0.1; https://doi.org/10.5281/zenodo.15122776).

The K30 communicates with a Boron microcontroller (Particle, San Francisco, CA, USA) over universal asynchronous receiver transmitter (UART) serial communication, which logs the readings to a micro-SD card on an Adalogger FeatherWing RTC + SD board (Adafruit, NYC, USA). All logged data can be downloaded from the micro-SD card by removing it from the Adalogger after deployment.

#### 2.1.3. Outer Housing

The entire interior system, including all its electrical components, was secured onto an electronics tray with zip ties that slides into a 4-inch diameter schedule-80 PVC tube, which was polished to be O-ring smooth (600 grit emery cloth) to ensure a waterproof seal with the O-ring flanges. On both ends of the PVC tube, there was a watertight end cap secured into place with an O-ring flange (Blue Robotics, Torrance, CA, USA). We included a serpentine channel on the 3D-printed end cap on the air side to increase the surface area of the exchanger. The end cap was designed to be compatible with the existing O-ring flange from Blue Robotics after being polished. The ePTFE membrane (IPE, Tempe, AZ, USA) was placed on top of the serpentine channel and secured in place using a 3D-printed retainer and 4–40 machine screws (Figure 1). To measure air temperature, a TMP117 protoboard (Adafruit, NYC, USA), which uses the TMP117 sensor from Texas Instruments (Dallas, TX, USA), was placed inside a custom 3D-printed housing that fits onto a Blue Robotics cable penetrator. The TMP117 is sealed into its housing using marine epoxy except for a small portion where the actual sensing component of the TMP117 is exposed, which was covered in thermally conductive epoxy in order to keep all elements waterproof.

For the water-side cap, we 3D printed a tubing holder that fits on top of the aluminum end cap from Blue Robotics. The 1.85 mm inner diameter ePTFE tubing (IPE, AZ, USA) was coiled around the 3D-printed tubing holder and connected to two cable penetrators potted with barbed tubing and sealed with marine epoxy, which connects to the rest of the airstream (Figure 1). We placed a temperature sensor (I2C Fast Response, Blue Robotics) on this side to measure water temperature. All 3D designs, including the K30 housing, were printed in Formlabs clear resin using a Formlabs Form 3 printer (Boston, MA, USA) and are available on the SEACOW Github repository.

### 2.2. Characterizing the SEACOW

#### 2.2.1. Air-Side Accuracy

Three additional SEACOWs were fabricated, and air-side measurements were compared between all four SEACOWs and those of a LI-850 (LI-COR Environmental, Lincoln, NE, USA). To assess their readings at a range of CO_2_ concentrations (0–1500 ppm), the rates of industrial-grade N_2_ and CO_2_ gas (Airgas, PA, USA) flowing into a sealed mixing chamber (~12 L) were varied, using calculations detailed in Appendix C and mass flow controllers. We increased the CO_2_ concentration by increments of ~250 ppm every 25 min. The air-tight mixing chamber was connected to the inputs and outputs of the LI-850 and the SEACOWs. Using the equilibrated readings from each SEACOW and the LI-850 at each step of the experiment, we produced dry calibrations (slope and intercept from a linear regression) for each SEACOW to fit the LI-850 data, which were then used in Equations (2)–(4), described below. The last 5 min of each step were deemed the equilibrated values. The LI-850 was calibrated prior to use according to instructions listed on LI-COR’s website [28].

#### 2.2.2. Response Time

The LI-850 has a 90% response time of <3.5 s, which, for our purposes and relative to the slower K30 and SEACOW, we considered to be effectively instantaneous. Therefore, it was used to indicate when the mixing chamber reached the goal concentration and to compare its readings to the SEACOWs’ response time. To estimate the air-side response time of the SEACOW, the mixing chamber was flooded with 1500 ppm CO_2_ until both the SEACOW and LI-850 stabilized. At a recorded time, the top of the box was removed and flooded with ambient air, facilitated with a fan. The time it took for each instrument to stabilize to the influx of ambient air was recorded, which provides an approximation of 5τ, or 99.3% response time, for the air side. Throughout the development process, we also repeated this process with several different potential diffusion membranes to evaluate their response times.

To estimate the 5τ time for the water side, a gas mixture of 1000 ppm CO_2_ was bubbled into 2 L of deionized water for 24 h to produce water with elevated pCO_2_. A stir plate and stir bar were used to ensure even mixing. Additionally, the gas mixture was bubbled first through a flask of deionized water prior to reaching the 2 L to humidify the gas stream and reduce evaporation. We placed the SEACOW, which had been sitting in air to equilibrate to the ambient room CO_2_ concentrations, into the 2 L of high pCO_2_ water and recorded the amount of time it took for the SEACOW to completely re-equilibrate as an estimate of the water-side 5τ time.

#### 2.2.3. Humidity and Pressure Correction

We used the humidity, pressure, and temperature measurements from the BME280 sensor inside the K30 housing, as well as the water/air-side temperatures, to correct our K30 readings following methods by Wall (2014), which we updated for our lower humidity air stream [17]:(2)$$K30_{CO2}=[(K30_{raw}-K30_{H2O})*m_{dry}]+b$$(3)$$K30_{H2O}={(m}_{H2O}*V_{H2O}+b_{H2O})*(H/100)$$(4)$$V_{H2O}=6*10^{-5}(T^{3})+5*10^{-4}(T^{2})+0.055(T)+0.571$$

In these equations, $K30_{CO2}$ is the corrected CO_2_ reading (µatm), $K30_{raw}$ is the reading before correction (µatm), $K30_{H2O}$ is the amount of the reading due to the presence of water vapor (µatm), $m_{dry}$ and *b* (µatm) are the slope and intercept of the dry calibration curve (as described in Section 2.2.1), $m_{H2O}$ (kPa µatm^−1^) and $b_{H2O}$ (µatm) are the slope and intercept of the K30 reading vs. vapor pressure curve linear regression, *H* (%) is the relative humidity inside the K30 housing, $V_{H2O}$ is the vapor pressure of water (kPa), and T (°C) is the water temperature. An example calculation using these equations can be found in the Appendix A.

To obtain the $m_{H2O}$ and $b_{H2O}$ first, a flask of deionized water was placed inside a water bath (6200 R20, Fisher Scientific, Waltham, MA, USA) through which we bubbled industrial-grade N_2_ gas to achieve a 100% humidity airstream, which fed into the K30 housing. The temperature of the water was varied from 18 to 24 °C to simulate a reasonable temperature range for the coast of North Carolina, and then the vapor pressure was calculated according to Equation (4). Once the vapor pressure values were calculated, a linear fit between vapor pressure and the K30 readings resulted in an $m_{H2O}$ and $b_{H2O}$ of −3.2 and 45 µatm, respectively. Note that no CO_2_ gas was used in this step, so the entire response is due to H_2_O.

This vapor pressure was calculated at a 100% humidity stream, but the inside the K30 housing only gets to 40–50% humidity during deployments due to the Nafion tubing and Drierite. Therefore, we multiply $K30_{H2O}$ by the proportion of the logged humidity to account for the fact that 100% humidity is not reached during actual deployments. We include the Drierite and Nafion tubing to ensure no moisture condenses on the internal electronics.

### 2.3. Laboratory Seagrass Experiment

A laboratory tank study was conducted in October 2023 to evaluate SEACOW’s ability to capture diel pCO_2_ cycling. Two tanks (one “experimental” and one “control”) were used as follows. For the experimental tank, approximately 0.65 m^2^ of *Halodule wrightii* was collected from a seagrass bed at 34.398° N, 77.616° W in September 2023 under UNCW CMS Collection DMF Permit #2037980 by coring the area to preserve the root system of the seagrass. Immediately following the coring, the seagrass and its sediment were placed into 4 rectangular plastic containers and placed into coolers, which were filled with seawater to avoid desiccation. Additionally, ≈45 L of sediment was collected from a seagrass-free area. The seagrass was placed in the experimental tank within 4 h of collection after its plastic container was further filled with the collected sediment to make sure it was securely planted. Sand (collected at 34.1934° N, 77.8047° W) was also sprinkled on top of the muddy sediment to decrease resuspension. The 95 L tank was filled with filtered seawater (<10 µm) to decrease the amount of biologically active material (i.e., living heterotrophs or autotrophs, which could influence CO_2_ and O_2_) in the tank. For the control tank, four more empty plastic containers were filled with the collected sediment and sand and placed in a different 95 L tank, which was also filled with the same filtered seawater. The experimental set up is pictured in Appendix A.

Each tank was outfitted with a hanging power filter (Qmax 90GPH), without the filter, to gently mix the water and a glass tank heater set to 22 °C, which is within the thermal optimal for *H. wrightii* [29]. Because the tanks received no additional water, they were prone to evaporation; accordingly, approximately 4 L of deionized water was added every 2–3 days to ensure the salinity of the tanks stayed consistently at 34–36 PSU. *H. wrightii* tolerates a wide range of salinities, from 25 to 45 PSU, with no changes in the growth rate; therefore, it was kept at an optimal range [29]. The tanks were uncovered. Four Finnex Planted+ (U-20WM; 23 W) aquarium lights were put on top of each tank to ensure sufficient lighting and were turned on from 6 am to 6 pm every day. One SEACOW and one dissolved oxygen (DO) sensor (miniDOT, PME, Vista, CA, USA) were placed in each tank. The SEACOW measured atmospheric pCO_2_ for 8 min and then switched to sampling aquatic pCO_2_ for 60 min at rate of 0.5 Hz. After sampling, it entered sleep mode for 52 min; consequently, one atmospheric and one aquatic pCO_2_ value is measured every two hours. The dissolved oxygen sensor sampled every 2 min continuously. We then averaged the last 5 and 3 min of raw data for each water-side and air-side cycle, respectively, which are the equilibrated end points reported in the results.

## 3. Results

### 3.1. Response Times of Different Membranes

During the development process, the response times of several different potential diffusion membranes were evaluated by flooding a chamber with a known concentration of CO_2_ and observing their response (Figure 3). The ePTFE (tubing 250 µm thick; planar 900 µm thick) had one of the fastest response times and higher durability than regular PTFE plumber’s tape (approx. 80 µm thick), so we chose it for our diffusion membranes. The silicone tubing we tested had a wall thickness of about 280 µm.

### 3.2. Air- and Water-Side Response Times

The averaged 99.3% air-side response time, or 5τ, of the SEACOW is 5.7 min, which we estimated by recording the amount of time it took for the SEACOWs to nearly equilibrate to the ambient air (Figure 4A). Therefore, 1τ is 1.14 min. It appears that there is not just a diffusion-limited response time but also a lag, likely due to “dead volume” in the system that needs to be flushed. It is important to note that the LI-850 did not have any diffusive barrier in place during this experiment and pumped at a rate of 0.75 L per minute. On the other hand, the SEACOW had the ePTFE planar diffusive membrane and pumps at a rate of about 0.20 L per minute. Note that SEACOW2 became inoperable due to a broken solenoid valve and was, therefore, excluded from the results. A more physically robust design to accommodate these or other solenoid valves could be introduced in the future design; however, it should be noted that we chose this particular solenoid valve because of its small size and low-power design.

The water-side response time (5τ) was calculated to be 29.6 min (Figure 4B). Only SEACOW3 was used during the water-side response time test due to spatial constraints in our tank. The equilibration of the water side is less stable than that of the air side, with a large spike occurring immediately after being placed in the water, which could be the result of the SEACOW being in close proximity to the operator’s breathing as it was placed. The general instability in the water-side response curve is likely caused by imperfections with our testing tank: the 2L container of DI water was open to the atmosphere while being bubbled with our gas mixture on a stir plate, potentially leading to small changes in aquatic pCO_2_ happening faster than the SEACOW could measure. Additionally, the diffusion of CO_2_ through water is slower than air, further contributing to the lag between the real time conditions and what the SEACOW was measuring. Additionally, the air stone used to bubble the 1000 µatm gas at a rate of 0.156 L min^−1^ into the water may not have been sufficient for reaching the goal concentration. While the goal was for the water to reach 1000 µatm in order to compare to the SEACOW’s reading for accuracy as well as response time, we cannot say for certain what the CO_2_ concentration of the water was without an independent measure and, therefore, used the data presented in Figure 4B to estimate the response time. Independent validation of water-side measurements would ideally occur through the comparison of aquatic SEACOW pCO_2_ measurements to aquatic pCO_2_ measured and/or estimated following best practices [30] and CO_2_ system calculations (e.g., [31]). However, unfortunately, we did not have access to high-quality bench-top inorganic carbon analyzers. Nonetheless, the SEACOW’s NDIR gas sensor, the K30, measures CO_2_ in the gas phase (Figure 2), and we thoroughly characterize gas-phase CO_2_ measurements through comparisons between the K30 and LI-850 (see Section 3.3 Air-side accuracy), thereby providing a whole-system accuracy benchmark.

### 3.3. Air-Side Accuracy

A stepwise gas experiment was conducted to assess variations in K30 (and therefore SEACOW) performance “out of the box” and after calibration. Both pre- and post-calibration, the SEACOWs had standard deviations < 3 µatm (the majority of which were <1 µatm), demonstrating the instruments’ stability across the full range sampled (Figure 5).

After the dry calibration constants were applied (Appendix D), SEACOWs reported stabilized measurements to within ~2.5% (as percent difference) of the LI-850’s readings on average, excluding values at 0 µatm (Figure 5). After calibration, the root mean square errors for SEACOWs 1, 3, and 4 in comparison with LI-850’s values are 9.97, 21.66, and 19.80 µatm, respectively. Water-side accuracy has not been separately characterized thus far, but given the use of the same K30 sensing unit for both water- and air-side measurements and the same diffusion membrane type, no significant difference is expected. Moreover, given that the final measurement is a ∆pCO_2_, any offset in the pCO_2_ calculated from the K30 response should be cancelled out. We acknowledge, however, that a difference in sensor gain (e.g., the slope from the dry calibration curves) could contribute to inaccuracy as the ∆pCO_2_ increases.

### 3.4. Seagrass Tank Experiment

SEACOWs collected ∆pCO_2_ data successfully in two tanks for two weeks. SEACOW1 was in the control tank, while SEACOW3 and SEACOW4 were in the seagrass tank. Values from SEACOW3 are excluded from Figure 6 due to reporting unreliable data, caused by an internal air leak visible as a crack in a 3D-printed tubing connector. The more robust design implemented 1/16” plastic barbed tube connectors (e.g., Part 5047K71 from McMaster-Carr, Douglasville, GA, USA). Dissolved oxygen data were especially noisy in the no-seagrass tank and are shown in the Appendix A. We suspect that small air bubbles may have been trapped on the surface of the DO sensor’s face. Both tanks (i.e., with and without seagrass) showed evidence of diel pCO_2_ variability, likely due to the presence of a microbial community in the sediment. However, pCO_2_ cycling is much more regular and amplified in the tank with seagrass (Figure 6), indicative of photosynthesis–respiration cycles. The lack of cycling in the DO readings in the no-seagrass tank also suggest that there was no consistent photosynthesis happening, presumably due to the lack of seagrass (Appendix A).

Throughout the experiment, the atmospheric pCO_2_ decreased during the day and rose at night (Figure 6). Aquatic pCO_2_ behaved as expected in the seagrass tank, with pCO_2_ decreasing during the day as photosynthesis occurred and increasing during the night as respiration happened. There is a large decrease in aquatic pCO_2_ on 16 October, which was due to the addition of deionized water to the tanks to maintain their salinity.

The difference between the averaged equilibrated water-side values and the air-side values was taken to produce ∆pCO_2_ values (Figure 7). From 11 October to 16 October, the ∆pCO_2_ values were as expected, with the seagrass tank seeing more consistent diel cycling, with values decreasing throughout the day and increasing through the night. Additionally, because the atmospheric pCO_2_ values from both SEACOWs matched closely (Figure 6), most of the differences between the ∆pCO_2_ values of each tank can be attributed to the aquatic pCO_2_.

One of the main results of this project is the creation of the SEACOW itself, for which its characterization data are summarized below:

Note that the accuracy in Table 1 was characterized in the range of 0 to 1500 µatm; therefore, the values reported here are only validated for CO_2_ concentrations below 1500 µatm. However, the K30 sensor used in the SEACOW has a sampling range of 0–5000 µatm.

## 4. Discussion

Sampling pCO_2_ in marine environments can be logistically difficult and expensive, yet it is a critical parameter to monitor as anthropogenic CO_2_ continues to change the ocean chemistry. Not only does monitoring pCO_2_ give scientists a better understanding of the quantity and effects of anthropogenic CO_2_ uptake, but it allows scientists to study many different aspects of oceanography, such as the movement of water along the oceanic conveyor belt [12], the effect pCO_2_ concentrations have on calcifying organisms [33], or the productivity of a region [34]. Additionally, there has been an increase in marine carbon dioxide removal (mCDR) groups who are looking for the best ways to monitor the movement of CO_2_, so they can verify their methods of sequestering CO_2_ from the atmosphere. Due to labor intensive and expensive methods of high-quality pCO_2_ sensing, many of these groups and scientists are interested in low-cost pCO_2_ technologies that can be deployed in larger quantities alongside more expensive, but singular, pCO_2_ systems. In addition to developing pCO_2_ monitoring for artificial carbon dioxide removal, it is important to monitor and protect blue carbon habitats that naturally sequester carbon, like seagrass meadows. Although we primarily discuss CO_2_ sensing of coastal and oceanic environments in this paper, expensive aquatic pCO_2_ sensors remain an issue for freshwater researchers as well, which has led to the development or application of similar instruments [24,35,36]. Furthermore, we acknowledge that the cost of the field instrument is comparably inexpensive, but some of the lab equipment needed for characterization (e.g., LI-850, gas tanks, 3D printer, and water bath) of this build increases the cost.

During deployments, we identified several areas of improvement for the SEACOW. Although the response time for the air side is sufficient for capturing rapid changes in atmospheric CO_2_ during deployments, the water-side response time is not as competitive when compared to other pCO_2_ instruments. The SIPCO_2_ (accuracy 29  ±  6 μatm) [16] and the CO2-Pro (accuracy ± 0.5%) (Pro-Oceanus) have reported approximate 5τ response times of 15 min and 12.5 min, respectively. The CO2-Pro’s efficient response time could be attributed to its pump that moves water across the equilibration membrane, which also reduces biofouling but increases its power consumption. The SEACOW’s water-side response time could be accelerated by adding an external pump or stirring mechanism, which may also assist in preventing biofouling. Additional research could also focus on improving the exchanger material and/or geometry (i.e., a higher surface area to volume ratio) as well as the rate of pumped air to assist with CO_2_ permeation [37]. We also note that Formlabs does not supply resin chemistry, and we, therefore, do not know what its CO_2_ permeability/absorptivity properties are for our 3D parts. However, given these results and the overall accuracy of the instrument, we do not consider this to be a significant source of uncertainty or response time lag. Additionally, we acknowledge that the lack of independent measurements of aquatic pCO_2_ during the water-side response time experiment results in uncertainty of the robustness of water-side measurements. In future developments of the SEACOW, resolving the water-side accuracy will be paramount. Additionally, we focus on data from one laboratory deployment in this paper (e.g., the seagrass experiment); therefore, it is imperative that future work focuses on replicating these types of experiments and performing other field deployments to assess the reliability of the system.

Finally, for the SEACOW to be used as an autonomous field instrument, the most pressing improvements for future deployments are the addition of a solar panel or larger battery pack, adding more Drierite to the system/using a different drying mechanism, and decreasing biofouling. When deployed in the field, the ePTFE tubing on the water side of the SEACOW became covered in algae after ~7 days; however, we found that rinsing the ePTFE tubing with deionized water from a wash bottle with a pointed nozzle was effective in removing the algae without damaging the ePTFE tubing. Currently, the Drierite is the biggest limitation for length of field deployments, becoming saturated after approximately 5 days of use. Depending on the application, the SEACOW could be modified to include a drying gas to extend this period. These advancements would allow the SEACOW to be deployed alongside more robust systems to increase spatial resolution of sampling. A possible design of a simple, low-cost float to deploy the SEACOW at the air–water interface is pictured in Appendix A. Additionally, the SEACOW may be secured to a floating dock for deployments.

## 5. Conclusions

In this project, we designed and built a ∆pCO_2_ device, characterized its functionality, and demonstrated its ability to capture the diel pCO_2_ cycling of seagrass in a controlled environment. The increasing popularity of microcontrollers and off-the-shelf parts help improve accessibility of monitoring technologies, which was one of the major goals of this project. We acknowledge that there are several improvements to be made on the SEACOW, and in the future, work will be focused on improving overall system accuracy characterization, decreasing the water-side response time, repeating characterization experiments and field deployments, and making the instrument more robust for longer field deployments. The work presented here further aids in characterizing the K30, which is increasingly popular in other low-cost atmospheric and/or aquatic CO_2_-sensing technologies. Because the SEACOW measures atmospheric and aquatic pCO_2_, a ∆pCO_2_ (pCO_2(water)_-pCO_2(air)_) value can be obtained and used to calculate CO_2_ fluxes. Therefore, even if the K30 sensor readings start to drift over time, the drift is subtracted out when the difference is taken, allowing the SEACOW to maintain rigor during deployments. This feature, along with the fact that the parts cost USD ~1400, make the SEACOW a valuable contribution to biogeochemical scientific and engineering communities.

## Figures and Tables

**Figure 1 sensors-25-03547-f001:**
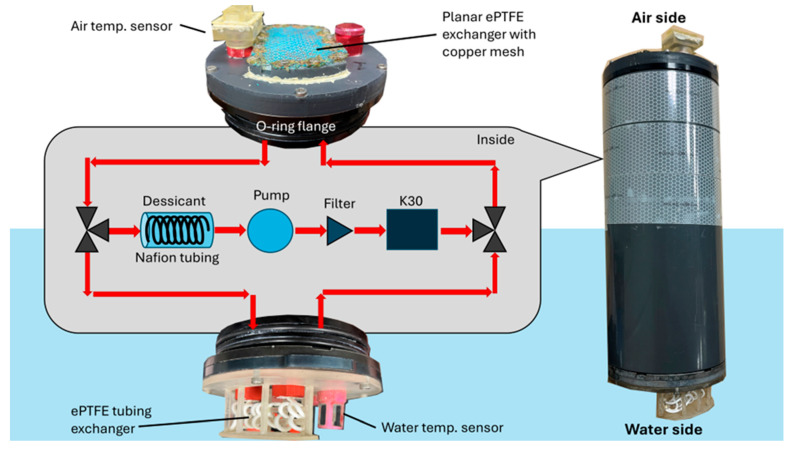
The plumbing diagram using ePTFE as a planar exchanger for the air side and ePTFE tubing for the water side. The red arrows indicate the direction of air flow. The components in the gray shaded region are the interior components of the device, the assembled version of which is shown on the right-hand side. The use of ePTFE as the exchanger was inspired by methods used in previous studies [24]. Once fully assembled, the SEACOW itself, shown in the right side of the figure, is 29.2 cm long with a diameter of 10.16 cm.

**Figure 2 sensors-25-03547-f002:**
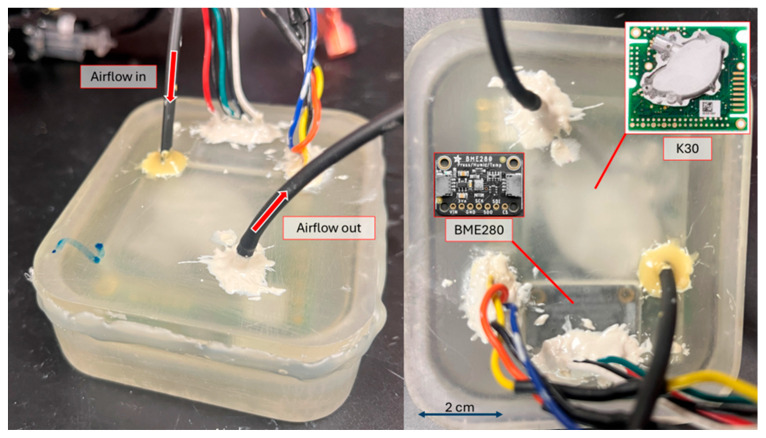
The 3D-printed K30 housing. The wire holes and top are sealed with marine epoxy to reduce gas leaks. It is 8.7 cm × 7.5 cm × 2.7 cm. The wires shown here are the communication and power to the BME280 and K30 sensors, which are housed inside. A scale is shown on the right-hand image.

**Figure 3 sensors-25-03547-f003:**
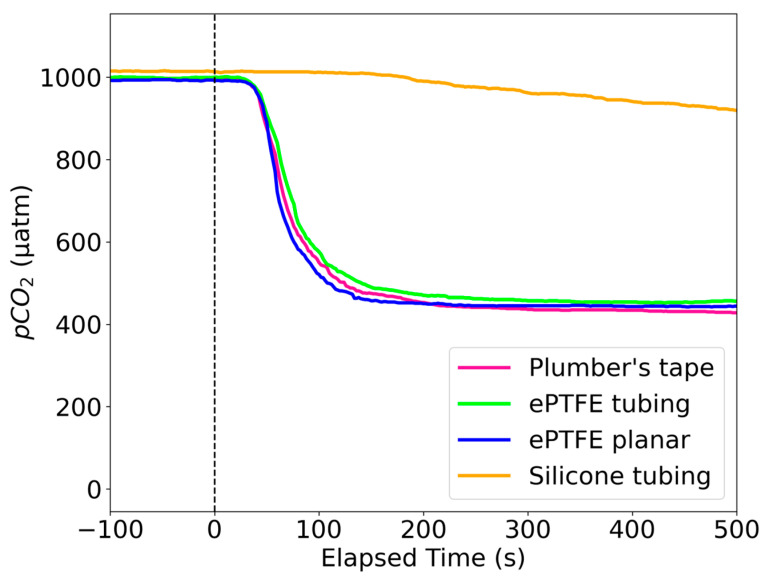
Response time of several semipermeable membranes to a change in pCO_2_ occurring at time 0.

**Figure 4 sensors-25-03547-f004:**
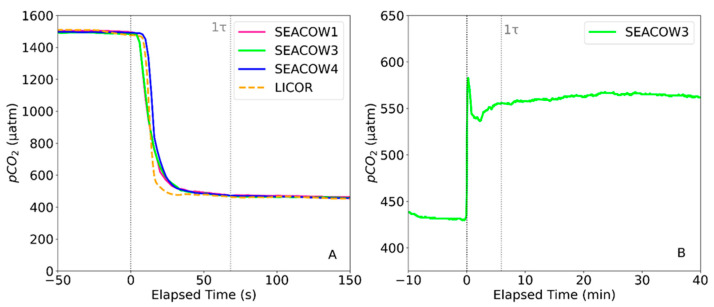
(**A**) The response of the SEACOWs air side to a step change in pCO_2_ occurring at time 0. Note that the x-axis has been limited to 150 s in order to emphasize the response within 1τ, as opposed to the full 5τ response. (**B**) The response of the SEACOW to a step change in aquatic pCO_2_ at time 0.

**Figure 5 sensors-25-03547-f005:**
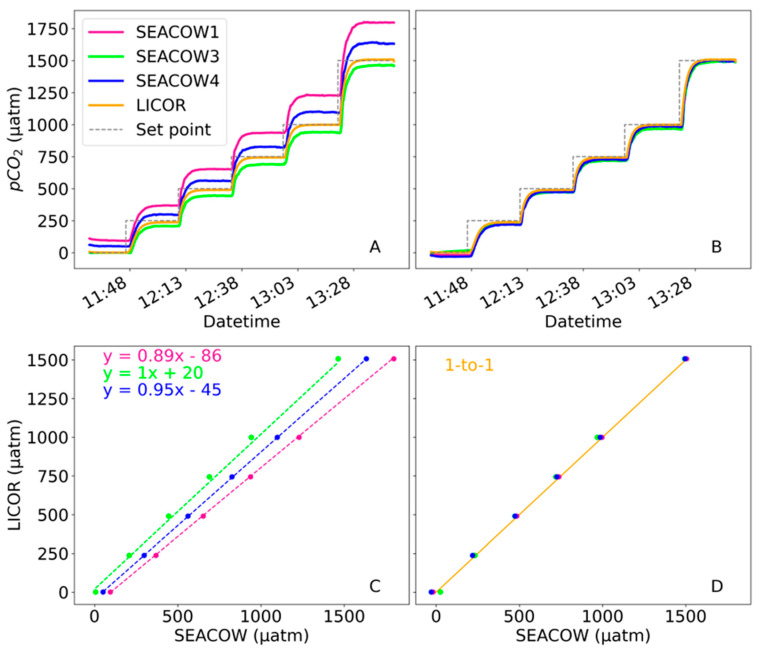
(**A**) Stepwise gas experiment of all four SEACOWs vs. the LI-COR LI-850 prior to applying the dry calibration curves. LI-COR LI-850 XCO_2_ measurements were converted to pCO_2_ for comparison. The dotted gray line represents the CO_2_ concentration set points throughout the experiment. (**B**) Stepwise gas experiment after applying the dry calibration curves. (**C**) The averages of the last 5 min of each ‘”step” plotted against their line of best fit, or the dry calibration curve, which are labeled using matching color. Error bars indicating standard deviation within the final five minutes were excluded from (**C**,**D**) due to being smaller than the markers and, therefore, not visible, so the data presented in these two panels can be found in table form in Appendix D. (**D**) The averages of the last 5 min of each “step” after applying the dry calibration curve. The orange 1-to-1 line is plotted using the data from LICOR averages as the x and y values.

**Figure 6 sensors-25-03547-f006:**
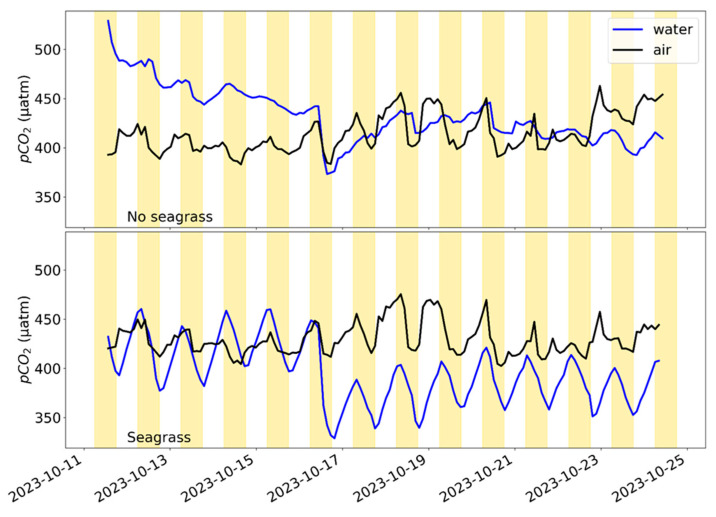
Equilibrated pCO_2_ end points plotted for the air (black) and water sides (blue) for the tanks with and without seagrass, as labeled. Yellow background signifies “daytime,” or when the aquarium lights are illuminated, and white background signifies “nighttime,” or when lights are turned off.

**Figure 7 sensors-25-03547-f007:**
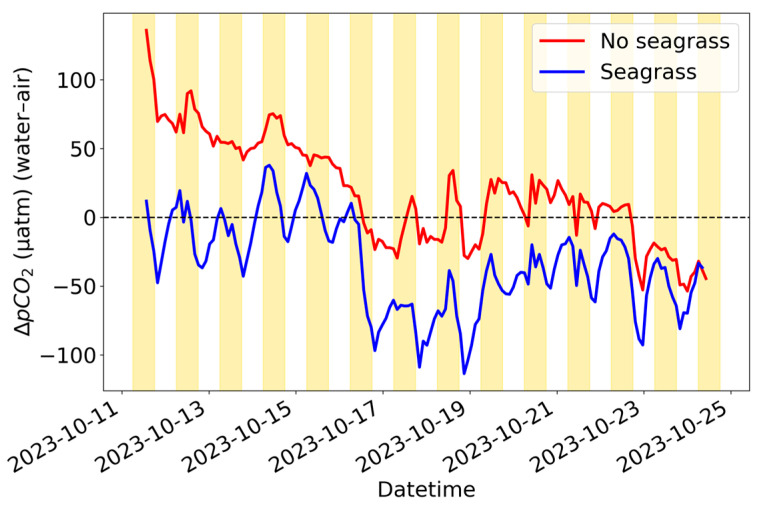
∆pCO_2_ (water–air) values for SEACOW1 and 4, which were in the control tank and seagrass tank, respectively.

**Table 1 sensors-25-03547-t001:** Summary of characterization data for the SEACOW.

Characterization Parameter	Value
Accuracy	±2.5% of LI-850’s readings
Air-side 5τ time	5.7 min
Water-side 5τ time	~30 min
Power draw	185 mW
Drierite budget	162 g per 5 days
Temperature range	5–40 °C
Cost in parts	~1400 USD
Github design files [32]	

## Data Availability

Design documents, including the 3D design files and circuit diagram, can be found on the SEACOW Github repository: https://doi.org/10.5281/zenodo.15122776.

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
