# Peer review of "Developing a Low-Cost Device for Estimating Air–Water ΔpCO_2_ in Coastal Environments"

_sensors, 2025, doi:10.3390/s25113547_

Round 1
Reviewer 1 Report
Comments and Suggestions for Authors
The study described in this manuscript describes a new affordable pCO2 sensor. As the authors state, and I concur, the dire need to pin down variability of air-to-water fluxes of CO2 is pressing but the very high costs of sensor technologies are holding progress back. In that respect, the authors’ work comes at a time of great need and any contribution is very welcome.
I reviewed this manuscript with the perspective of a potential user who has only fundamental understanding of the electronics and sensing background needed to design and produce this instrument. The manuscript is generally well-written with accurate and clear phrases, adequately referenced without being overwhelming, and accompanied by suitable graphs of good quality. I particularly appreciated the authors’ candor on all aspects of sensor development and testing, e.g., the detection of an operator’s breath when setting up a response-time experiment as a spike in the pCO2 record. My overall evaluation is that the sensor the authors describe is promising and I would gladly consider using it myself in my own research.
Given the above, I’d like to recommend that the manuscript be published after some important revisions (and several minor ones) which I provide to the authors next:
- Your manuscript is replete with very important details about the SEACOW itself, especially section 2.1 and Appendix A. Details on powering the device and communicating with it for data download are scarce and actually left to the imagination of the reader. Please consider adding a section 2.1.3 that discussed what happens beyond the SEACOW itself.
- The cost of the SEACOW in parts is mentioned a few times throughout the manuscript. However, it is obvious that calibration is absolutely necessary and an integral part in making a newly made device functional. Am I mistaken? If not, you should devote a dedicated paragraph or two in describing the additional equipment needed to make it functional, including a robust lab sensor like the LI-850, gases, water baths, etc. Perhaps this could be a small subsection at the end of the Results section, called “SEACOW manufacture and calibration summary” or something along those lines, that could also include Table 2 which now seems out of place.
- In your abstract you mention that you “provide recommendations for SEACOW field deployments” but these were hard to find. The final Discussion section paragraphs (lines 401-428) should be expanded to cover important aspects that are currently missing. Here’s a short (and non-exhaustive) list:
- How would you recommend maintaining the SEACOW at the air-water interface, e.g., deploy it within a float? A short mention would be useful.
- Does the current design of the underside that is in contact with water easy to clean periodically (at least once a week in a Carolina summer) from biofouling organisms the way other field gear is, e.g. using brushes, dental tools, etc.? In my experience, this is necessary even with instruments that have a wiper brush system, like YSI’s EXO2s.
- Assuming a sampling frequency of 15 or 20 minutes, given the 5t value you report, how long would a deployment last given the current battery supply?
- Are the issues your prototypes 2 (solenoid valve) and 3 (crack in a tubing connector) a cause of concern, suggesting alternative products that may be necessary? By the way, I applaud your transparency and I commend you for it. I’m trying to be constructively critical. :)
- Please provide units following the description of each variable in Equations 2-4 (lines 207-212).
- On the data in Table 1:
- I’d recommend converting Table 1 to a 3-panel figure, with every panel dedicated to each SEACOW’s data, overlaying the three curve formulas in each. Table 1 could be included in the Supplementary materials document. The standard deviations are so low that they will be smaller than the symbol size, so your point in the text about their magnitude will be well made.
- You may consider adding a fourth panel with the LICOR data on the x-axis and the “Post” data of the three SEACOWs on the y-axis, which will dovetail nicely with figure 5b.
- Please check the best-fit line formulas again. I plotted the SEACOW1 data and while I also got a slope of 0.89, I found an intercept of -101.
- The following are minor editorial-style comments:
- Line 42: Change “they” to “the authors”
- Line 53: Change “uptakes” to “uptake”
- Line 112: Place “±3% of the reading” in parentheses (?)
- Line 122: Remove extra space between “over” and “universal”
- Line 127: Error message “Error! Reference source not found.” Missing citation?
- Lines 209-210: Change phrase in parentheses to “(as described in Sect. 2.2.1)”
- Line 348 onwards: Move this paragraph before Figure 6 for convention (figure follows the text it’s cited in)
- Line 369: Start this sentence with “The difference…” and cite Figure 7 at the end of the sentence.
- Line 397: Place the “2” in CO2 in subscript.
- Line 417: Place the “2” in CO2 in subscript.
- Figures S2 and S3: Please increase the font size of the text (formula and R2 value) in the figure area, or add this information in the caption for Figure S2 (you already do so in Figure S3).
- In Supplementary materials: There is mention of Figure S4 on p. 3, but no Figure S4. Include this figure or remove its mention.
Author Response
Thank you very much for taking the time to review this manuscript and providing detailed and constructive feedback. We appreciate your helpful comments and suggestions. Please find the detailed responses below and the corresponding revisions in track changes in the re-submitted files. Note that the line numbers we provide in our responses below correspond to the line numbers in the final document without tracked changes.
|
Comments 1: Your manuscript is replete with very important details about the SEACOW itself, especially section 2.1 and Appendix A. Details on powering the device and communicating with it for data download are scarce and actually left to the imagination of the reader. Please consider adding a section 2.1.3 that discussed what happens beyond the SEACOW itself. |
|
Response 1: Thank you for pointing this out. We agree and therefore have created a new section (Section 2.1.2 Electronics and software) to augment clarifying information to the existing text to address your comment, line 116. |
|
Comments 2: The cost of the SEACOW in parts is mentioned a few times throughout the manuscript. However, it is obvious that calibration is absolutely necessary and an integral part in making a newly made device functional. Am I mistaken? If not, you should devote a dedicated paragraph or two in describing the additional equipment needed to make it functional, including a robust lab sensor like the LI-850, gases, water baths, etc. Perhaps this could be a small subsection at the end of the Results section, called “SEACOW manufacture and calibration summary” or something along those lines, that could also include Table 2 which now seems out of place. Response 2: We agree that it should be mentioned that the SEACOW cost does not include the other instruments needed to characterize it; therefore, we have added a sentence to the Discussion to further explain this: “Furthermore, we acknowledge that the cost of the field instrument is comparably inex-pensive, but some of the lab equipment needed for characterization (e.g., LI-850, gas tanks, 3D printer, water bath) of this build increase the cost.” Additionally, the use of the LI-850 and gases is described in Methods Section 2.2.1 “Air-side accuracy” and the use of the water bath in Methods Section 2.2.3 “Humidity and pressure correction.” |
|
Comments 3: In your abstract you mention that you “provide recommendations for SEACOW field deployments” but these were hard to find. The final Discussion section paragraphs (lines 401-428) should be expanded to cover important aspects that are currently missing. Here’s a short (and non-exhaustive) list:
Response 3: We agree that the field recommendations could be more robust, and have added our responses below in the order of your bullet points: · We added two sentences in the Discussion, copied below, to address how we deployed the SEACOWs at the air-sea interface, and we added a photo of an example float to the supplemental information (Figure S5): “A possible design of a simple, low-cost float to deploy the SEACOW at the air-water interface is pictured in Figure S5. Additionally, the SEACOW may be secured to a floating dock for deployments.” · We added a sentence to describe how we cleaned the biofouling on the water-side of the SEACOW: “When deployed in the field, the ePTFE tubing on the water-side of the SEACOW became covered in algae after ~7 days; however, we found that rinsing the ePTFE tubing with deionized water from a wash bottle with a pointed nozzle was effective in removing the algae without damaging the ePTFE tubing.” · Given the 5τ reported here (~ 30 minutes for the water-side), we believe the sampling interval used is appropriate and do not think it fits in this article to consider the wide-ranging possibilities of other sampling intervals; however, to address the reviewer’s question, we have included the power draw which, along with a battery’s capacity, can be used by anyone to estimate deployment lifetimes. · We thank the reviewer for this constructive criticism. Accordingly, in lines 305-308, we added a sentence to why we chose this solenoid valve and added a sentence in lines 370-373 to address why we don’t believe the cracked tubing connector is a cause for concern for the SEACOW. Comments 4: Please provide units following the description of each variable in Equations 2-4 (lines 207-212). Response 4: Thank you, we have added the units accordingly and updated the text in lines 226-231. Comments 5: On the data in Table 1:
Response 5: Thank you for your thorough consideration of optimal data presentation and providing these suggestions. As suggested, we moved Table 1 into the Appendix, and we remade Figure 5 to include two additional panels to show the data from Table 1, including the lines of best fit. We added in the manuscript where the table could be found and noted in the figure caption that the standard deviations were excluded from Figure 5, as they were too small to be seen. With respect to your slope and intercept, we double-checked all of our best-fit line formulas and confirmed they are correct using the data from Table 1. In doing so, we also caught one typo in the SEACOW3 formula, whose intercept should be 20 instead of 40. These changes can be found in Appendix C and Figure 5. Comments 6: · The following are minor editorial-style comments: · Line 42: Change “they” to “the authors” · Line 53: Change “uptakes” to “uptake” · Line 112: Place “±3% of the reading” in parentheses (?) · Line 122: Remove extra space between “over” and “universal” · Line 127: Error message “Error! Reference source not found.” Missing citation? · Lines 209-210: Change phrase in parentheses to “(as described in Sect. 2.2.1)” · Line 348 onwards: Move this paragraph before Figure 6 for convention (figure follows the text it’s cited in) · Line 369: Start this sentence with “The difference…” and cite Figure 7 at the end of the sentence. · Line 397: Place the “2” in CO2 in subscript. · Line 417: Place the “2” in CO2 in subscript. · Figures S2 and S3: Please increase the font size of the text (formula and R2 value) in the figure area, or add this information in the caption for Figure S2 (you already do so in Figure S3). · In Supplementary materials: There is mention of Figure S4 on p. 3, but no Figure S4. Include this figure or remove its mention. Response 6: Thank you for the detailed comments and suggestions; we have made all the changes you suggested. |
Reviewer 2 Report
Comments and Suggestions for Authors
Minor:
- line 66: Why were the square brackets used here? Explain better the term kw
- line 101: "24 in of nafion tubing" please use metric units
- line 108: |11.5 inches long with a diameter of 4 inches" please use metric units. Make these changes throughout the text.
- line 127: Fix the missing reference here, and throughout the text.
- line 142: "which was polished to be o-ring smooth" Explain better why was this performed.
- line 205: Provide reference for V_H20 formula and coefficients used
- line 229: provide a photograph of the experimental setup in section 2.3
Major:
- Improve quality and explanation of Figure 1, and explain the direction of red arrows in the figure. Discuss the cases of partial pressure of CO2 being larger in water than in air and vice versa. Explain better the function of 3-way valves. Improve schematics on the left hand side to better match the photo of the device shown on the right hand side.
- Improve the quality of Figure 2, so that the size of the 3D printed box can be directly compared to the rest of the developed instrument.
- Explanation of Fig 4b is not satisfactory. Both the description of experimental setup and the results should be improved.
- In line 416 authors acknowledge that "Additionally, we acknowledge that the lack of independent measurements of aquatic pCO2 during the water-side response time experiment results in uncertainty of the robustness of water-side measurements." Indeed the manuscript has two sections devoted to the Air-side accuracy (2.2.1 and 3.3). The difficulties that prevented accessing water side accuracy should be elaborated, and possible solution (even if not implemented in this manuscript) should be discussed.
Author Response
|
1. Summary |
|
|
|
Thank you very much for taking the time to review this manuscript and for providing feedback. Please find the detailed responses below and the corresponding revisions/corrections highlighted/in track changes in the re-submitted files. Note that the line numbers we provide in our responses below correspond to the line numbers in the final document without tracked changes. |
||
|
3. Point-by-point response to Comments and Suggestions for Authors |
||
|
Comments 1: Improve quality and explanation of Figure 1, and explain the direction of red arrows in the figure. Discuss the cases of partial pressure of CO2 being larger in water than in air and vice versa. Explain better the function of 3-way valves. Improve schematics on the left hand side to better match the photo of the device shown on the right hand side.
|
||
|
Response 1: Thank you for your comments. For Figure 1, we changed the word “Drierite” to dessicant and added a bubble to clarify that the schematics on the lefthand side are happening inside the device on the right side. Additionally, we added more information in the figure caption to explain the direction of the red arrows and to reiterate that the components in the gray shaded region are the insides of the device. Furthermore, we added a sentence in lines 103-105 to better describe the function of the solenoid valves. |
||
|
Comments 2: Improve the quality of Figure 2, so that the size of the 3D printed box can be directly compared to the rest of the developed instrument. Response 2: Thank you for the suggestion. To clarify the size of the K30 housing, we have added a scale bar on the right-hand side of Figure 2. Additionally, the dimensions of the K30 housing and the SEACOW as a whole can be found in the captions of Figure 2 and Figure 1, respectively. |
||
|
Comments 3: Explanation of Fig 4b is not satisfactory. Both the description of experimental setup and the results should be improved. Response 3: We thank you for your comment. We have added a clarifying phrase in lines 212-215 o further describe the process of obtaining the data shown in Figure 4B.
Comments 4: In line 416 authors acknowledge that "Additionally, we acknowledge that the lack of independent measurements of aquatic pCO2 during the water-side response time experiment results in uncertainty of the robustness of water-side measurements." Indeed the manuscript has two sections devoted to the Air-side accuracy (2.2.1 and 3.3). The difficulties that prevented accessing water side accuracy should be elaborated, and possible solution (even if not implemented in this manuscript) should be discussed. Response 4: We agree with your comment; therefore, we have added a sentence in lines 332-334 to address the lack of water-side accuracy data in this manuscript. Comments 5: Minor: Response 5: Following your suggestions, we have made the following changes: - line 66: Changed the square brackets to parentheses - line 101: Changed 24 in to metric units - line 112: Changed dimensions of the SEACOW to metric units - line 137: fixed the missing reference here - line 158-159: added a clarifying phrase to describe why we did this “…to ensure a waterproof seal with the o-ring flanges.” - line 220-224: The V_H20 formula and coefficients used are cited as: Wall (2014), or reference 17. - line 229: A photograph of the experimental setup for the seagrass experiment has been included in the supplemental material as Figure S4, which we’ve noted in line 265. |
||
Reviewer 3 Report
Comments and Suggestions for Authors
This manuscript presents the development and validation of SEACOW, a low-cost, IoT-enabled ΔpCO₂ sensor system capable of measuring both atmospheric and aquatic pCO₂. It offers a valuable contribution to improving carbon flux observations at the air–sea interface. My minor comments are as follows:
(1) (4,127) Please correct the erroneous reference. Since this section describes the design, does it refer to findings from previous studies? If so, please clarify and cite appropriately.
(2) (5,167) The manuscript states that the SEACOW system was calibrated against the LI-850, which measures CO₂ concentrations in the range of 0 to 1500 ppm. Does this imply that SEACOW is only validated for CO₂ concentrations below 1500 ppm? If so, how would the device perform under significantly higher CO₂ levels (e.g., ~10,000 ppm)? Please clarify the measurement range and any limitations in sensor accuracy or calibration at elevated concentrations.
(3) (11, 348) Due to the malfunction of SEACOW2 and SEACOW3, only one valid dataset is available for each tank (with and without seagrass). How do the authors justify the reliability of the system and present meaningful error bars with such limited replication? Please clarify how the performance and efficiency of the SEACOW system are validated under these conditions.
Author Response
|
1. Summary |
|
|
|
Thank you very much for taking the time to review this manuscript and providing feedback. We appreciate your comments. Please find the detailed responses below and the corresponding revisions/corrections highlighted/in track changes in the re-submitted files. Note that the line numbers we provide in our responses below correspond to the line numbers in the final document without tracked changes.
|
||
|
|
|
|
|
3. Point-by-point response to Comments and Suggestions for Authors |
||
|
Comments 1: (1) (4,127) Please correct the erroneous reference. Since this section describes the design, does it refer to findings from previous studies? If so, please clarify and cite appropriately. |
||
|
Response 1: Thank you for pointing this out. It was an internal cross-reference to Figure 2, and it has been fixed. |
||
|
Comments 2: (2) (5,167) The manuscript states that the SEACOW system was calibrated against the LI-850, which measures CO₂ concentrations in the range of 0 to 1500 ppm. Does this imply that SEACOW is only validated for CO₂ concentrations below 1500 ppm? If so, how would the device perform under significantly higher CO₂ levels (e.g., ~10,000 ppm)? Please clarify the measurement range and any limitations in sensor accuracy or calibration at elevated concentrations. Response 2: We appreciate your comment. Yes, the SEACOW is only validated for CO2 concentrations below 1500 ppm; however, the K30 sensor used within the SEACOW has a measurement range only up to 5000 ppm. We have added a sentence after Table 1 in lines 409-412 to explicitly state this. |
||
|
Comments 3: (3) (11, 348) Due to the malfunction of SEACOW2 and SEACOW3, only one valid dataset is available for each tank (with and without seagrass). How do the authors justify the reliability of the system and present meaningful error bars with such limited replication? Please clarify how the performance and efficiency of the SEACOW system are validated under these conditions. Response 3: We agree that there were numerous challenges due to the limited deployments of this proof-of-concept design, and we attempt to address several areas where we see potential for improvement, especially in the last two paragraphs of the Discussion. We added a sentence in lines 454-457 to acknowledge this challenge and the future work needed. We believe the robustness of results indicated in the stepwise gas experiments (including post-calibration accuracy and high repeatability) further demonstrate the capabilities of the technology. |
||
Reviewer 4 Report
Comments and Suggestions for Authors
Oceans cover the majority (71%) of our planet and are a very important element of our ecosystem. Ocean vegetation and organisms have a very strong influence on the concentration of gases in the atmospheric air, including the concentration of CO2. Monitoring and analyzing changes in the exchange of gases between atmospheric air and water is crucial in understanding the phenomena taking place and in the decision-making process aimed at protecting the ecosystem of our planet. Increasing the number of measurement points significantly affects the quality and accuracy (resolution) of the observations made. For these reasons, I consider the article to be necessary and timely.
The article is properly divided into sections, the introduction sufficiently describes the current state of knowledge and presents the direction taken by the authors of the manuscript. In relation to general comments, I have one surprising one, namely that, as a person for whom English is not a native language used on a daily basis, it was difficult for me to read the text of the manuscript. I could not find the meaning of the words used in the text (for example "diel", "in atm"). This is surprising because, looking at the authors' affiliation, one would expect the highest quality of English. Perhaps these are colloquial expressions used in the authors' scientific community, or my knowledge of vocabulary was simply insufficient.
I also have also a few additional particular comments to the content of the presented article. The order of the comments does not reflect their significance. It results only from the order of appearance in the text of the article.
My remarks and comments:
1. Lines 22-23, “After calibration, the SEACOW reports atmospheric pCO2 measurements within 2–3% of measurements made by a calibrated LI-COR LI-850.” - incomprehensible sentence, did the authors mean uncertainty here?
2. Line 24, “diel” - what does this word mean?
3. Line 70, “atm” - what does this word mean?
4. Lines 112-113, “which can be improved significantly with calibration” - low cost sensors are usually characterized by high short- and long-term variability of static characteristics. This is due to the low quality of the materials used as well as the simplified production process. It would be very valuable to describe how a sensor with poor metrological parameters can be transformed into a sensor with better properties. Calibration only allows for a temporary adjustment of the readings, nothing more.
5. Line 125, “3D-printed housing” - it would be interesting for the reader to know what material was used? The choice of material is not as simple as one might expect.
6. Lines 127-128, “placed a BME280 Sensor (Adafruit, NYC, USA)” - The BME280 sensor is manufactured in Germany by Bosch Sensortec GmbH, Reutlingen
7. Line 149, “TMP117 sensor (Adafruit, NYC USA)” - the sensor is manufactured by Texas Instrument USA; I recommend accurately assigning elements to manufacturers, not to companies that used the element to create the measurement module. The measurement characteristic is related to the element, not the measurement module.
8. Line 190, “a gas mixture of 1000 ppm CO2” - what was the carrier gas? what was the composition of the gas mixture?
9. Line 215, “pure N2 gas” - what was the purity of the gas. Technical gas with a purity of 3.5 may contain about 500 ppm of other gases, for example carbon dioxide.
10. Line 254, “aquarium lights” - what was the range of the spectrum of the emitted light, did it correspond to the spectrum in the place where the water grass occurred, it would also be interesting for other scientists what was the power of the light sources or what was the radiation flux.
11. Line 314, “typically < 1 μatm” - this designation is rather unfortunate; if the standard deviation < 3, then the probability of error < 1 is 0.2586. Is this really a typical value?
12. Table 2, accuracy - I suggest presenting it as a value, not in relation to another device.
13. Table 2, “drierite” - what does this word mean?
14. Line 413, “formlabs” - what does this word mean?
15. Line 502, “Diaphragm gas pump (UNMP 05)” - if previously the elements were specified together with the manufacturer, then in the case of this pump KNF Germany should be added, especially since it is a high quality pump often used in gas analyzers. This is valuable information for other scientists.
Author Response
|
Thank you very much for taking the time to review this manuscript and providing feedback. We appreciate your comments. Please find the detailed responses below and the corresponding revisions/corrections highlighted/in track changes in the re-submitted files. Note that the line numbers we provide in our responses below correspond to the line numbers in the final document without tracked changes. |
||
|
3. Point-by-point response to Comments and Suggestions for Authors |
||
|
Comments 1: Lines 22-23, “After calibration, the SEACOW reports atmospheric pCO2 measurements within 2–3% of measurements made by a calibrated LI-COR LI-850.” - incomprehensible sentence, did the authors mean uncertainty here? |
||
|
Response 1: Thank you for letting us know. We have reworded the sentence in lines 22-23 to be more clear. Additionally, we define the value as percent difference in line 344.
|
||
|
Comments 2: Line 24, “diel” - what does this word mean? Response 2: The word “diel,” or daily, is very commonly used in the natural and physical sciences and we leave it here without further definition, especially for this work submitted to the “Environmental Sensing” section. |
||
|
Comments 3: Line 70, “atm” - what does this word mean? Response 3: We have added the definition of “atm,” or atmosphere, to the “Abbreviations used in this text” section at the end of the manuscript. Comments 4: Lines 112-113, “which can be improved significantly with calibration” - low cost sensors are usually characterized by high short- and long-term variability of static characteristics. This is due to the low quality of the materials used as well as the simplified production process. It would be very valuable to describe how a sensor with poor metrological parameters can be transformed into a sensor with better properties. Calibration only allows for a temporary adjustment of the readings, nothing more. Response 4: We agree with the reviewer that low-cost sensors have their flaws, especially surrounding drift and accuracy. Accordingly, we identify and discuss several areas of improvement for the SEACOW throughout the Discussion section, especially lines 436-472. Comments 5: Response 5: We describe the material used in line 178. Comments 6: Response 6: Thank you for pointing this out. We have added a clarifying sentence in lines 140-141. Comments 7: Response 7: We agree with your suggestion and have added clarifying information in lines 166-167. Comments 8: Response 8: We describe the gas mixture (N2 and CO2) in lines 185-187 and provide calculations for our mixture in Appendix B. Comments 9: Response 9: We changed “pure” to “industrial grade” to describe the purity of the gas in Line 235. Comments 10: Response 10: We have added the aquarium lights model number and wattage in line 274, so other researchers may be able to find this information if needed. Comments 11: Response 11: We have reworded the sentence in line 339-340 to confirm that the standard deviation was <1 uatm for the majority of all reported SEACOW readings in Table 1. Comments 12: Response 12: While we appreciate your suggestion and see your point about this comment, we have to present the accuracy in reference to the LI-850 since that is the metric by which we compared to. If we do not include it, the reader may incorrectly assume how the accuracy information was derived. Comments 13: Response 13: We define drierite as a desiccant in line 101. Comments 14: Response 14: We define Formlabs as the company from which we bought our 3D printer and resin from in lines 177-179. Comments 15: Response 15: We agree. All part names, manufacturers, and links to components can be found in the Bill of Materials on the public SEACOW Github page, which is noted in lines 143-144. |
Round 2
Reviewer 2 Report
Comments and Suggestions for Authors
Authors have adressed majority of issues raised in previous round, however, I would like them to elaborate a bit more on:
Minor comment related to previous comment 1: Discuss the cases of partial pressure of CO2 being larger in water than in air and vice versa.
Minor comment related to previous comment 4: The difficulties that prevented accessing water side accuracy should be elaborated. To what extent is this water side accuracy adressed in the current version of manuscript?
Author Response
|
1. Summary |
|
|
|
Thank you very much for taking the time to review our responses to reviewers and providing feedback. Please find the detailed responses below and the corresponding revisions highlighted in tracked changes in the re-submitted files. Also, note the line numbers we refer to below correspond to the line numbers in the .pdf version of the submitted manuscript. |
||
|
3. Point-by-point response to Comments and Suggestions for Authors |
||
|
· Comments 1: Minor comment related to previous comment 1: Discuss the cases of partial pressure of CO2 being larger in water than in air and vice versa. |
||
|
Response 1: Thank you for the comment. In our interpretation of this comment, we believe that the necessary information can be found in Equation 1 and the subsequent definition of ∆pCO2 which defines the chosen sign convention (pCO2water – pCO2air) and therefore describes that when pCO2water > pCO2air, the flux, F, will be positive, or, when pCO2water < pCO2air, F will be negative. However, we have added a sentence in lines 69-70 to explicitly state this. Additionally, we mention some of the causes in changes of CO2 concentrations in lines 38-42. |
||
|
· Comments 2: Minor comment related to previous comment 4: The difficulties that prevented accessing water side accuracy should be elaborated. To what extent is this water side accuracy adressed in the current version of manuscript? Response 2: Toward this point, we further elaborate on our description of water-side accuracy, our ability to evaluate it, and future proposed work in lines 332-339. The text addressing this point now reads, “Independent validation of water-side measurements would ideally occur through the comparison of aquatic SEACOW pCO2 measurements to aquatic pCO2 measured and/or estimated following best practices (Dickson 2007) and CO2 system calculations (e.g., van Heuven et al. 2011). However, unfortunately, we did not have access to high-quality bench-top inorganic carbon analyzers. Nonetheless, the SEACOW’s NDIR gas sensor, the K30, measures CO2 in the gas phase (Figure 2) and we thoroughly characterize gas-phase CO2 measurements through comparisons between the K30 and LI-850 (see 3.3 Air-side accuracy), thereby providing a whole-system accuracy benchmark.” Furthermore, we recommend the following in Section 4. Discussion in Lines 455-459: “Additionally, we acknowledge that the lack of independent measurements of aquatic pCO2 during the water-side response time experiment results in uncertainty of the robustness of water-side measurements. In future development of the SEACOW, resolving the water-side accuracy will be paramount.” |
||